# Effects of Morphology on the Bulk Density of Instant Whole Milk Powder

**DOI:** 10.3390/foods9081024

**Published:** 2020-07-31

**Authors:** Haohan Ding, Bing Li, Irina Boiarkina, David I. Wilson, Wei Yu, Brent R. Young

**Affiliations:** 1Department of Chemical & Materials Engineering, University of Auckland, Auckland 1010, New Zealand; hdin307@aucklanduni.ac.nz (H.D.); bing.li@auckland.ac.nz (B.L.); b.young@auckland.ac.nz (B.R.Y.); 2Fonterra Co-Operative Group Limited, Auckland 1010, New Zealand; Irina.Boiarkina@fonterra.com; 3Electrical and Electronic Engineering Department, Auckland University of Technology, Auckland 1010, New Zealand; david.wilson@aut.ac.nz

**Keywords:** instant whole milk powder, bulk density, morphology, principal component analysis, partial least squares, artificial neural networks

## Abstract

The chemical and physical properties of instant whole milk powder (IWMP), such as morphology, protein content, and particle size, can affect its functionality and performance. Bulk density, which directly determines the packing cost and transportation cost of milk powder, is one of the most important functional properties of IWMP, and it is mainly affected by physical properties, e.g., morphology and particle size. This work quantified the relationship between morphology and bulk density of IWMP and developed a predictive model of bulk density for IWMP. To obtain milk powder samples with different particle size fractions, IWMP samples of four different brands were sieved into three different particle size range groups, before using the simplex-centroid design (SCD) method to remix the milk powder samples. The bulk densities of these remixed milk powder samples were then measured by tap testing, and the particles’ shape factors were extracted by light microscopy and image processing. The number of variables was decreased by principal component analysis and partial least squares models and artificial neural network models were built to predict the bulk density of IWMP. It was found that different brands of IWMP have different morphology, and the bulk density trends versus the shape factor changes were similar for the different particle size range groups. Finally, prediction models for bulk density were developed by using the shape factors and particle size range fractions of the IWMP samples. The good results of these models proved that predicting the bulk density of IWMP by using shape factors and particle size range fractions is achievable and could be used as a model for online model-based process monitoring.

## 1. Introduction

Bulk density, also called packing density, represents the weight of powder per unit volume, and could generally be stated in kg/m^3^ or g/cm^3^ [1,2]. Bulk density is vital for instant whole milk powder (IWMP) because it affects the packing, transportation, and processing of IWMP, all of which affects the profit of IWMP processing [3]. For example, when transporting milk powder over long distances, high bulk density can reduce the cost of shipping and packaging materials, while when selling milk powder, low bulk density can make a milk powder more competitive than other brands’ higher density milk powder because it has larger volume per given weight [2]. Furthermore, bulk density is an important parameter describing the flowability of milk powder [4]. Since agglomeration increases the particle size, it significantly reduces the bulk density of milk powder [2,5]. Therefore, the bulk density of instant whole milk powder is generally lower than the bulk density of regular whole milk powder. Additionally, since free-flowing property is improved by agglomeration [2], regular whole milk powder always has a poorer flowability compared with agglomerated/instant whole powder.

Generally, there are four kinds of bulk density measured: compact bulk density, tapped bulk density, loose bulk density, and aerated bulk density [3]. Since the relationship between tapped and loose bulk density is an effective evaluation of cohesion, tapped and loose bulk density are the most basic and useful terms in describing powder behaviour [6]. Tuohy [7] reported that the tapped bulk density of whole milk powder, skim milk powder, and fat-filled milk powder varied between 0.44 to 0.85 g/cm^3^. At present, multiple methods can measure the bulk density of milk powders [2]. Nijdam and Langrish [8] used a graduated cylinder of 1 g of milk powder, then used the change of volume after tapping to calculate the loose and tapped bulk densities, while Pugliese et al. [1] used a 100 mL calibrated cylinder of milk powder, before using the change of weight to calculate the loose and tapped bulk densities. However, all current bulk density measurement methods are post-production tests. The shortage of real-time feedback makes the real time control of bulk density difficult. Additionally, bulk density is also crucial for some other powders. For example, it is vital for pharmaceutical powders since it determines the complexity of handling, storing and processing these powders [9]. What is more, bulk density is also an important physical parameter for soil because it can estimate its water-related characteristics [10].

Many factors determine the bulk density of milk powder. For example, the relative humidity and the drying temperature can affect milk powder’s bulk density [6,8]. Additionally, Pisecky [2] pointed out that the density of solids, the amount of interstitial air, and the sphericity of the particles determine the bulk density of milk powder. Furthermore, Pisecky [2] also indicated that interstitial air and particle shape are the dominating factors determining the bulk density of agglomerated powders, while the bulk density of non-agglomerated powders is controlled by particle size distribution. Since the non-agglomerated particles are recycled from the spray dryer chamber and the rest are removed in the fluid bed, IWMP mostly consists of agglomerated particles [2,11]. Therefore, the IWMP’s bulk density is determined by the particle shape. Similarly, Bhandari [5] indicated that the agglomeration process influences the bulk density, and Abdullah and Geldart [6] pointed out that particle shape affects the aerated and tapped bulk density of powders. However, most papers qualitatively use scanning electron microscopy (SEM), to visually analyze the morphology of powders. For example, Gaiani et al. [12] used SEM to study the surface differences between native phospho-caseinate powders and native phosphor-caseinate powders with addition of lactose and ultrafiltrate, respectively, but they did not find any visual differences between the powders. SEM was also used by Murrieta-Pazos et al. [13] to investigate the morphological differences between regular whole milk powders and skim milk powders, and it was found that the surface of whole milk powders is smooth and homogeneous while the surface of skim milk powders is wrinkled. In these studies, morphological characteristics have not often been the key focus but rather have merely been supporting information, and the qualitative descriptions used could generally only be identified in powders with significant compositional differences such as skim, whole, and instant whole milk powders.

This work explores a quantitative connection between the morphology and bulk density of a single type of milk powder (IWMP) with the ultimate aim of achieving control of bulk density. To obtain milk powder samples with various particle size fractions, different brands of IWMPs were sieved into three particle size groups and then recombined by using the simplex-centroid design. The quantitative information on different shape characteristics of the powders was gathered by light microscopy and image processing, and the tapped and loose bulk density of each milk powder sample were measured by tap testing. In order to confirm which shape factor mostly affects the bulk density of IWMP, principal component analysis was used. Regression analyses using partial least squares (PLS) and artificial neural networks (ANN) were used to assess the effectiveness of bulk density prediction using the morphology of IWMP.

## 2. Experiment

### 2.1. Experimental Procedure

#### 2.1.1. Powder Preparation

Since it is difficult to divide milk powder into different groups by morphology, and different industrial plants may produce milk powders which have different bulk density and morphology, different brands of IWMP samples, denoted as Brands 1, 2, 3 and 4, were purchased. All of these milk powders were bought off the shelf in a local Auckland, NZ, supermarket. Additionally, although some other properties may affect the bulk density of IWMP, e.g., moisture and oil content, the bulk density of IWMP is mainly determined by particle shape and the amount of interstitial air [2]. The aim of this work is to develop on-line or at-line sensors to predict the bulk density of IWMP; the effects of other properties on the bulk density of IWMP will be studied in the future.

#### 2.1.2. Sieving

In previous research, the particle size groups had different diameter ranges [2,3,14] and the prediction performance of the model might have been affected by the range of the diameters. In this research, a Retsch AS200 sieve shaker was used to divide the milk powders into different but consistent particle size range groups: coarse particles (>355 μm), medium particles (180–355 μm) and fine particles (<180 μm). 150 g samples of milk powder were shaken for 30 min before the powders were transferred to foil bags to ensure stable moisture content. To minimize the breakdown of agglomerates, the sieve shaker used was a vibratory type and the amplitude selected was low. It is worth mentioning that the milk powder samples that were measured came from a packer and had already gone through significant dense phase vacuum transport. Therefore, only the most robust powder is left behind at this point, as everything else is crushed during transport [15]. Finally, the milk powders from different particle size fractions were remixed following a simplex centroid design, discussed in Section 2.1.5.

#### 2.1.3. Image Processing

Particle shape factors were obtained by light microscopy and image processing. Firstly, 0.08 g IWMP was weighed out before dispersing into 16 mL canola oil, followed by stirring at 600 rpm for 10 s [14]. 1 mL of the oil and powder combination was then loaded by a pipette onto a clean microscope slide. The photos of IWMP particles were taken by a light microscope which consisted of a ×10 objective lens and a ×10 Moticam camera. For each slide, twenty photos were taken, and for repeatability three slides were analysed for each sample. An example image of the agglomerated particles is presented in Figure 1a. Lastly, a custom MATLAB function was used to process these images. The boundaries of each milk powder particle were computed and extracted from images, as shown in Figure 1b. Although there are some alternative microscopy techniques that can obtain particle shape information, such as stereo microscope and confocal microscope, we used light microscopy since it is cheap and easy to operate, and the best choice for at-line sensor development [11].

#### 2.1.4. Tap Testing

The bulk density of IWMP was obtained by tap testing. This test uses a Quantachrome Autotap that is equipped with a moving platform and a hollow circular box whose mass and volume are known. A clear plastic cylinder sits over this box that allows excess powder to be poured over the volume limit. Powder was poured into the box, and excess powder removed by a flat ruler, before weighing the total mass of powder and box to calculate the loose bulk density by Equation (1):(1)$$D_{L}=\frac{M_{l}-M_{c}}{V_{c}}$$
where *D_L_* is the loose bulk density, *M_c_* represents the mass of the box, *M_l_* represents the total mass of powder and box before tapping, and *V_c_* is the volume of the box.

Next, a clear plastic cylinder was fixed on the container, and more milk powder was poured into this cylinder. The container was then tightened onto the auto tap machine. After tapping 1250 times [2,16], the plastic cylinder was removed, and the excess powder was again removed by the flat ruler. Finally, the container was weighed, and Equation (2) was used to compute the tapped bulk density of IWMP:(2)$$D_{T}=\frac{M_{t}-M_{c}}{V_{c}}$$
where *D_T_* is the tapped bulk density and *M_t_* represents the total mass of powder and container after tapping.

#### 2.1.5. Simplex-Centroid Design (SCD)

As the proportion of the component is determined by a mixture, simplex-centroid design (SCD) can examine the relationship between a mixture component and response variables [17,18]. In this research, the boundaries of the design factors were determined by the boundaries of components [19]. The boundaries of the components were controlled by sieving results: fine particles occupied 20–40% of the IWMP, medium particles were 50–70% of the IWMP, and the boundaries of coarse particle fraction ranged from 10% to 30%. Figure 2 shows the particle size fractions of the IWMP samples. For example, in sample 6, coarse particles account for 10%, 70% of the particles are medium, and fine particles account for 20%. Firstly, the milk powders were divided into pure coarse particles (the first sample), pure medium particles (the second sample), and pure fine particles (the third sample). The milk powders were then remixed by using the SCD to get the other seven samples. As a result, each brand of milk powder has ten samples in total.

### 2.2. Morphology Analysis

#### 2.2.1. Shape Factors

The boundaries of each milk powder particle in the images that were extracted by the custom MATLAB functions were used to compute the shape factors of each IWMP particle. There are many 2-D shape factors [20], and generally one or two shape factors were chosen to describe the shape of particles [21]. In this research, the shape factors used to describe milk powder particles were: 2-D cross-sectional area, perimeter, equivalent diameter, elongation, solidity, convexity, circularity, and maximum and minimum Feret diameters. The definitions and detailed explanations of these shape factors were summarized by Ding et al. [11]. Figure 3 shows the detailed information on these shape factors.

#### 2.2.2. Statistical Analysis and Mathematical Modelling

##### Principal Component Analysis (PCA)

PCA is a useful technique to obtain important information from a data set that consists of many interrelated dependent variables and uses a reduced set of new orthogonal variables (principal components) to represent this important information [22,23,24]. It has been widely used in various scientific fields [25].

In this research, ten points (10% to 100% percentiles in increments of 10%) from an empirical cumulative distribution function (ECDF) for each shape factor value (maximum Feret diameter, circularity, elongation, etc.) were selected to construct the models. The variable size (90 variables) is much bigger than the sample size (40 samples), which means that the matrix for PCA analysis is ill conditioned [26]. Therefore, three shape factors (maximum Feret diameter, circularity, and elongation) were chosen by PCA to develop models. However, the variable size, which has 30 shape factor variables, was still very big. So PCA was used again to reduce the number of shape factor variables. In the end nine ECDF points were chosen, and the variable size is smaller than the sample size. These shape factor variables were then used to develop the PLS and ANN models.

##### Partial Least Squares (PLS) Regression

PLS regression is a powerful technique of analysis because of the small limitation on variable size, noise, and sample scales [11]. In particle analysis, PLS has been widely used [27,28,29] since it can confirm the relationships between independent variables and dependent variables [30]. In this research, PLS models were developed to predict the bulk density of IWMP by using shape factors and particle size fractions. To avoid model overfitting, cross-validation was used to determine the number of PLS components [31]. The samples were separated into five groups, and the *R*^2^ (multiple correlation coefficient) and *Q*^2^ (the cross-validated *R*^2^) of the model were calculated as follows [11]:(3)$$R^{2}=1-\frac{RSS}{SS}$$
(4)$$Q^{2}=1-\frac{PRESS}{SS}$$
where *RSS* is the sum of the squares of the fitted residuals, *SS* is the sum of the squares of the difference between actual Y values and its mean values, and *PRESS* is the sum of squares of the differences between the actual and predicted Y values for the selected data.

##### Artificial Neural Networks (ANNs)

An ANN is a multiple parallel computing system which contains a number of neurons in layers which can fit the data while interconnecting with each other [32,33]. ANNs are very prevalent in classification, prediction, optimization, and clustering because of their excellent performance in processing nonlinear signals [34]. In this research, MATLAB was used to solve the fitting problem. The network of the neural net fitting application was a two-layer feed-forward network which included sigmoid hidden neurons and linear output neurons because this feed-forward neural network model can use consistent data, and has enough neurons in its hidden layer to fit the multi-dimensional mapping problems arbitrarily well. Additionally, the samples were divided into three subsets: training, validation, and testing. The training subset accounts for 70%, the validation subset occupies 15%, while the rest of the data were used for testing. After testing a different number of hidden neurons, it was found that ten hidden neurons performed the best. The Levenberg-Marquardt algorithm which requires more memory but less time was chosen to train the model and to avoid overfitting [35]. When the mean squared error (MSE) of the validation samples stopped reducing, the training automatically ended.

## 3. Results and Discussion

### 3.1. Univariate Analysis

To study the link between morphology and bulk density of IWMP, the morphology of each brand’s milk powders was compared. The ECDF values of the selected shape factors (maximum Feret diameter, circularity, and elongation) from PCA results, were used to describe the morphology of each milk powder sample. Table 1 lists the bulk densities of each brand’s coarse, medium, and fine particle samples. The mean value and standard deviation of three repeated experiment were computed. From Table 1, it is clear that all coarse particle samples have the lowest bulk density while all fine particle samples show the highest bulk density. The main reason is that coarse particles are more irregular than medium particles and fine particles, while the fine particles are the most spherical [11], and the spherical shaped particles will lead to higher bulk density caused by the low interstitial air content, while the irregular shaped particles will cause lower bulk density [2].

#### 3.1.1. Fine Particles

Figure 4 shows the ECDF of maximum Feret diameter, circularity, and elongation for fine particle samples from four different brands of IWMP. It is clear that the ECDF curves of elongation and circularity for Brand 3 are higher than other brands, and the tapped and loose bulk density of Brand 3′s fine particles are significantly lower. In addition, the ECDF curves of elongation and circularity for Brand 1 are lower than other brands, and the tapped and loose bulk density of Brand 1′s fine particles are higher. Furthermore, the values of elongation and circularity of Brands 2 and 4 are similar, and the tapped and loose bulk density of Brands 2 and 4 are also similar, indicating that circularity and elongation can distinguish high bulk density from low bulk density in fine particle size fractions of IWMP. But the maximum Feret diameter of four different brands is similar, therefore, the bulk density of IWMP cannot be distinguished by maximum Feret diameter from the fine particle size fraction.

The mean values of maximum Feret diameter, elongation, and circularity for fine particles of four different brands of IWMP versus their tapped and loose bulk density values are presented in Figure 5. As circularity and elongation increase, the bulk density also increases, while as maximum Feret diameter increases, the bulk density decreases. This may be because the milk powders with more regular particle shape will cause low interstitial air content, which will lead to a high bulk density. Figure 5 also shows that the circularity, elongation, and bulk density of Brands 2 and 4 are very similar.

#### 3.1.2. Medium Particles

The ECDFs of maximum Feret diameter, elongation, and circularity are illustrated in Figure 6. It is noticeable that the ECDF curves of elongation and circularity for Brand 1 are well separated from the other three brands, and the tapped and loose bulk density of Brand 1′s medium sized particles are also significantly higher in comparison. Additionally, the values of circularity and elongation of Brand 3 is the highest while the loose bulk density of Brand 3 is the lowest. Consequently, in the medium particle size group, the loose bulk density of IWMP can be differentiated by elongation and circularity. However, the tapped bulk density of Brand 3 is higher than that for Brands 2 and 4, so circularity and elongation cannot distinguish the tapped bulk density of IWMP in the medium particle size group. Furthermore, since the values of maximum Feret diameter of Brands 1, 2, 3, 4 are similar, the maximum Feret diameter also cannot distinguish the bulk density of IWMP for the medium particle size fraction.

Figure 7 shows the mean values of shape factors (maximum Feret diameter, elongation, and circularity) for medium particles of different brands versus their tapped and loose bulk density. Similarly to the fine particles, with a decrease in maximum Feret diameter and an increase in circularity and elongation (more regular particle shape) the loose bulk density shows an overall upward trend (low interstitial air content). Additionally, with a decline in maximum Feret diameter, the tapped bulk density rises, but the trend of tapped bulk density versus the change of circularity and elongation is unclear.

#### 3.1.3. Coarse Particles

Figure 8 presents the ECDF of maximum Feret diameter, elongation, and circularity for coarse particle samples from four different brands of IWMP. The maximum Feret diameter of Brand 1 has the highest ECDF curve, while the maximum Feret diameter of Brand 4 has the lowest ECDF curve, and the bulk densities of Brand 1 are the highest, while Brand 4′s milk powders have the lowest tapped and loose bulk density. Additionally, the ECDF values of circularity for Brand 1 are significantly lower than the other three brands, while the bulk densities of Brand 1 are much higher than the other three brands. Therefore, in the coarse particle size group, the tapped and loose bulk density of IWMP can be distinguished by maximum Feret diameter and circularity. However, due to the similar ECDF curves of elongation for four different brands, elongation cannot differentiate the bulk density in the coarse particle size fraction.

The mean values of maximum Feret diameter, elongation, and circularity of four different brands’ coarse particles versus their tapped and loose bulk density are illustrated in Figure 9. It is clear that as maximum Feret diameter increases and the circularity decreases (more irregular particle shape), the bulk density decreases (high interstitial air content). It is also notable that even though the brand order for highest to lowest bulk density has changed from what was seen with fine particles, the trends in tapped and loose bulk density have remained consistent with the particle circularity and maximum Feret diameter. Additionally, when the elongation of coarse particles is around 0.61, the bulk density of IWMP coarse particles is maximum.

In conclusion, for the fine-sized particles, elongation and circularity can differentiate high bulk density from low bulk density for the IWMP, but the maximum Feret diameter cannot. In the medium-sized particle group, elongation and circularity can distinguish the loose bulk density of the IWMP. However, they cannot differentiate the tapped bulk density, while the maximum Feret diameter still cannot distinguish the tapped and loose bulk density of the IWMP. For the coarse-sized particles, the maximum Feret diameter and circularity can differentiate the tapped and loose bulk density of the IWMP, but elongation cannot. Additionally, for all the particle size fractions (fine particles, medium sized particles, and coarse particles), as the maximum Feret diameter decreases or the circularity increases, the bulk density of the IWMP increases, which may be because the milk powders with small particle size and high circularity (more regular) will lead to low interstitial air content, which will cause a high bulk density. However, with an increase in elongation, there is no clear trend for tapped and loose bulk density. Furthermore, although the moisture and oil content may affect the bulk density of IWMP [5], these are controlled very tightly during milk powder production [2], so the difference of moisture and oil content between four different brands’ IWMP should not be appreciable.

### 3.2. Principal Component Analysis

To decide which shape factor dominates the bulk density of IWMP, PCA was used. Figure 10a shows the scores plot of PCA, in which variables 1 to 12 represent the four different brands’ coarse particle, medium particle, and fine particle fractions. Since the coarse particle samples have the lowest tapped and loose bulk density, and the fine particle samples have the highest tapped and loose bulk density, therefore the samples were well classified by their bulk density along the axes in the PCA scores plot. It is also clear that the first two principal components were enough to categorize the bulk density of the IWMP because they account for 88% and 6% of the total variance, respectively. Figure 10b presents the PCA loading plot. It is noticeable that the shape factors were divided into two groups. One of them includes all the size metrics like minimum and maximum Feret diameter, equivalent diameter, perimeter, and area. In contrast, the other group contains all the shape metrics (circularity, solidity, convexity, and elongation). As in the same group the shape factors have similar loadings, the maximum Feret diameter, elongation, and circularity, for which loadings were relatively high, were chosen to train models to predict the bulk density of IWMP.

For each shape factor, there are ten variables (shape factor_0.1_ to shape factor_1.0_, where the ECDF fraction is represented by a subscript). Hence, the data set has 30 shape factor variables and 12 samples. Since the variable size is large, PCA was used to determine which shape factor variables most significantly contribute to the bulk density of the IWMP. Figure 11a shows the PCA scores plot. It is clear that PC1 and PC2 explain 75% and 10% respectively of the complete variance, and the milk powder samples were well classified with their bulk density. The PCA loadings plot is presented in Figure 11b, where the points branded 1 to 30 are the 30 shape factor variables (maximum Feret diameter_0.1_ to maximum Feret diameter_1.0_, circularity_0.1_ to circularity_1.0_, and elongation_0.1_ to elongation_1.0_). Since the loadings of some shape factor variables are similar, nine variables with relatively high loadings were chosen to represent the shape factors of IWMP. Therefore, points 3 (maximum Feret diameter_0.3_), 7 (maximum Feret diameter_0.7_), 10 (maximum Feret diameter_1.0_), 13 (circularity_0.3_), 17 (circularity_0.7_), 19 (circularity_0.9_), 22 (elongation_0.2_), 23 (elongation_0.3_), and 27 (elongation_0.7_) were chosen to represent the shape factors of the IWMP. From the shape factors’ ECDF curves, it is clear that at the selected points, the differences between shape factors are considerable between the different brands’ milk powders.

### 3.3. Prediction of the Bulk Density

Each brand of IWMP was remixed into ten samples with varying fractions by using the simplex-centroid design. Next, tap testing was used to measure the bulk density of each sample. Table 1 shows the tapped and loose bulk density of the first three samples for each brand, while the bulk densities of the other seven samples for each brand are presented in Table 2. The proportion of the different sized particles in each milk powder sample was determined by the simplex-centroid design in Section 2.1.5. All experiments were repeated three times. The mean value with the standard deviation of bulk density is listed in Table 2. Each milk powder sample has three particle size groups, and there are nine shape factor variables for each particle size group. Therefore, there are 27 shape factor variables for each sample. To predict the bulk density of the IWMP, the shape factor variables of each particle size group were multiplied by their fraction to get the composite values, which were used for the PLS and ANN analysis.

#### 3.3.1. Partial Least Squares Models

Two PLS models were developed by using the shape factor variables selected in Section 3.2 to predict the tapped and loose bulk density of IWMP, respectively. The results of the first PLS model (for predicting loose bulk density) are shown in Figure 12a,b. Since the milk powder samples were separated into five groups, there are eight samples in each group. The *R*^2^ and *Q*^2^ of the model were computed by Equations (3) and (4). Figure 12a presents the change of *R*^2^ and *Q*^2^ with the growing number of PLS components, and when the PLS model has four components, the *Q*^2^ reached its maximum value (0.87), and the corresponding *R*^2^ is 0.94. In this condition, the PLS regression of the actual loose bulk density versus the predicted loose bulk density is shown in Figure 12b, and the MSE of the model is 65.51.

The second PLS model was constructed for predicting the tapped bulk density of IWMP. Figure 13a shows the change of the *R*^2^ and *Q*^2^ of the model with an increasing number of PLS components. A parsimonious solution was chosen when the number of PLS components is small (two) and *Q*^2^ is near maximum. For this, the *R*^2^ value is 0.92, and the *Q*^2^ is 0.91. The predicted tapped bulk density versus actual tapped bulk density is presented in Figure 13b, and the MSE of the model is 227.49. The PLS models are considered good as the *R*^2^ of the models is high (0.94 and 0.92), and the mean squared errors are reasonable.

#### 3.3.2. Artificial Neural Network models

ANN models were also constructed for predicting the bulk density of IWMP. The shape factor variables selected from PCA results were multiplied by their particle size fractions and introduced into the ANN models. The first ANN model was constructed for predicting the loose bulk density of IWMP. The validation of this ANN model is shown in Figure 14a where the MSE of the validation model reached its minimum value for an epoch number of four. Therefore, four epochs were selected. Figure 14b presents the performance of the first model, and Table 3 presents the results of this ANN model. These results are discussed below.

The second ANN model was constructed for the tapped bulk density of the IWMP. The validation of the second ANN model is shown in Figure 15a, where we can see that when the epoch number was two, the MSE of the validation model reached its minimum value. So, two epochs were used. Figure 15b shows the performance of this ANN model, and Table 4 presents the results of this model.

These two ANN models are considered good since the *R*^2^ (0.98 and 0.98) of the models are high, and the mean squared errors (22.02 and 63.99) are low. Since the performance of the ANN models is better than the performance of the PLS models here, on this basis ANN could be considered as potentially being useful for at-line prediction. However, due to the black-box characteristic of the ANN models, the mechanism behind an ANN is rarely understood. Contrary to this, PLS ban be more easily interpreted to reflect fundamental changes that could be understood by operators and process engineers. Therefore, PLS is recommended for use in the current industry [36].

In summary, the bulk density of IWMP mainly depends on powder morphology and particle size. Furthermore, since these models only depend on the particle size fraction and shape factors of the milk powder but are independent of the brand of the milk powder, using these models to predict the bulk density of IWMP is applicable. Compared to traditional time-consuming and labor-intensive off-line bulk density testing, using the PLS and ANN models developed in this work is potentially helpful to build on-line or at-line sensors, which is necessary for industry 4.0. Additionally, Depree et al. [11] used process variables (pressure, flow, temperature, etc.) and online quality variables (moisture, fat, protein, etc.) to develop a partial least squares model for predicting the bulk density of IWMP. However, the precision (*R*^2^ is lower than 0.8) of using process variables to predict bulk density of IWMP is much poorer than the precision (*R*^2^ is about 0.94) of using morphology, which may be because the bulk density of the agglomerated powders is mainly determined by particle shape [2]. Furthermore, 3D image processing technologies which can obtain more accurate information may be applied to these smart sensors in the future.

## 4. Conclusions

By investigating the relationship between morphology/particle size and bulk density, the current research aims to develop an on-line or at-line sensor for bulk density measurement. The proposed methodology can improve plant efficiency and reduce operating costs as an alternative to the traditional bulk density test. It was found that the shape factors (maximum Feret diameter, elongation, and circularity) for different brands of IWMP are different. For different particle size groups, the trends of the tapped and loose bulk density versus the changes of the shape factors (maximum Feret diameter, elongation, and circularity) are similar. In addition, PCA could well classify the bulk density of IWMP by their shape factors. Finally, two PLS models and two ANN models were developed using selected shape factor variables to predict the tapped and loose bulk density of IWMP. The good results of these models indicate that using the shape factors and particle size fractions of IWMP to predict the bulk density of IWMP is applicable. Due to the black box limitations of ANN models, PLS models are considered more useful to operators for indicating trends, although ANN models performed marginally better than the PLS models.

## Figures and Tables

**Figure 1 foods-09-01024-f001:**
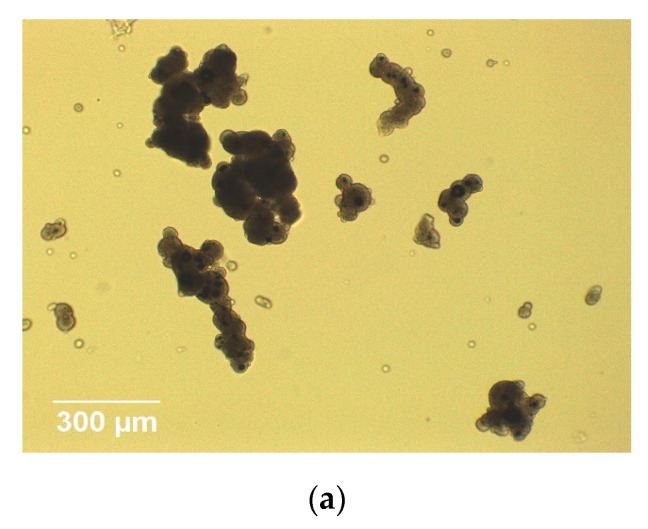
(**a**) Image of agglomerated milk powder particles; (**b**) Particle identification processed by MATLAB.

**Figure 2 foods-09-01024-f002:**
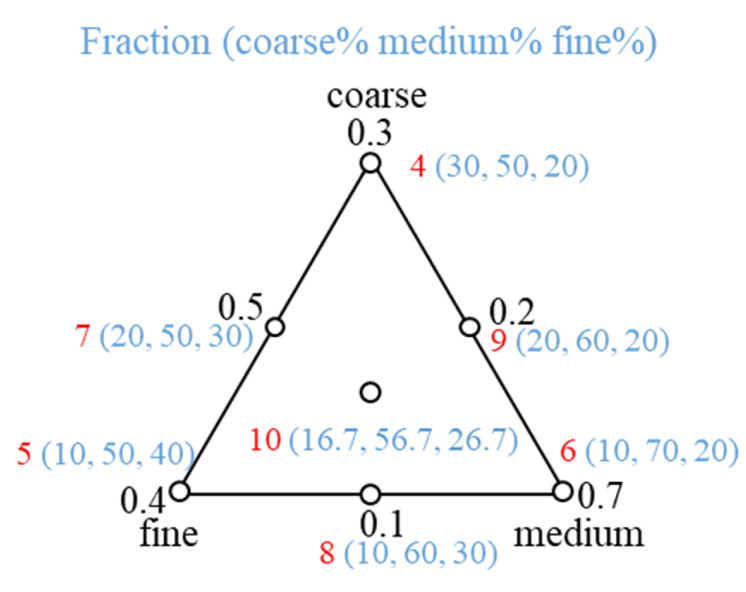
Different particle size fraction choices based on simplex-centroid design.

**Figure 3 foods-09-01024-f003:**
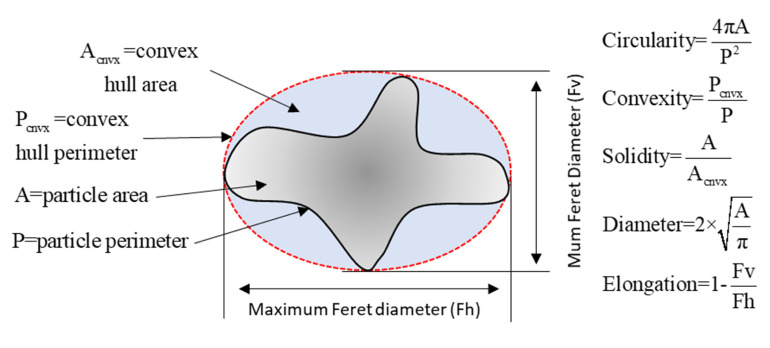
Shape factors that are used for model construction.

**Figure 4 foods-09-01024-f004:**
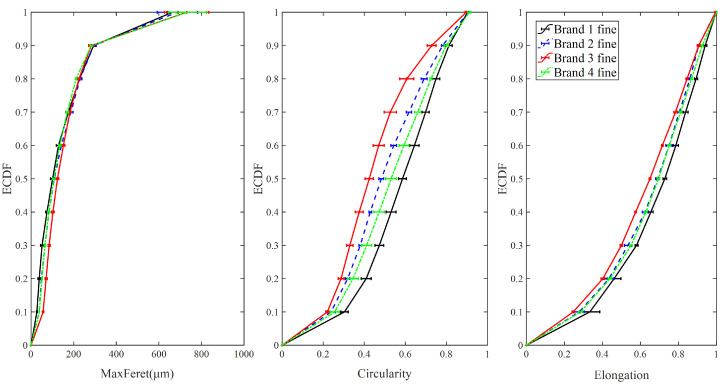
The empirical cumulative distribution function (ECDF) of shape factors for fine particles.

**Figure 5 foods-09-01024-f005:**
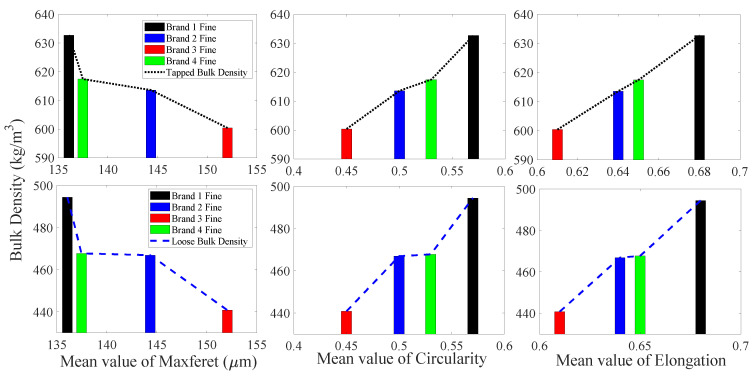
Tapped and loose bulk density from four different brands of instant whole milk powder versus mean values of shape factors for fine particles.

**Figure 6 foods-09-01024-f006:**
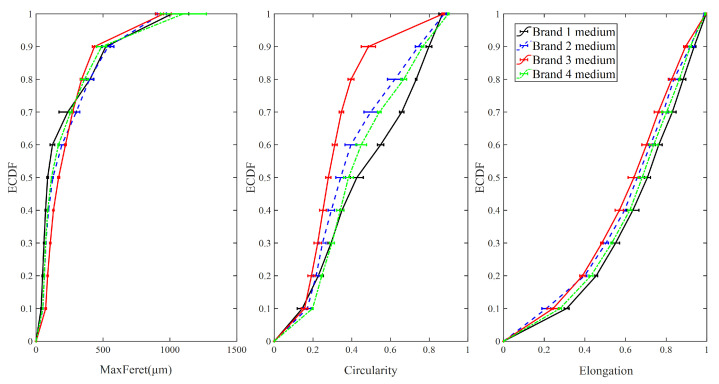
The empirical cumulative distribution function of shape factors for medium sized particles from four different brands of instant whole milk powder (IWMP).

**Figure 7 foods-09-01024-f007:**
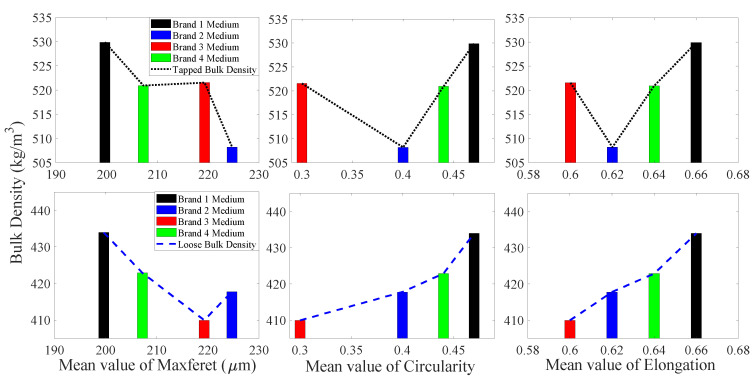
Tapped and loose bulk density of four different brands of instant whole milk powder versus mean values of shape factors for medium sized particles.

**Figure 8 foods-09-01024-f008:**
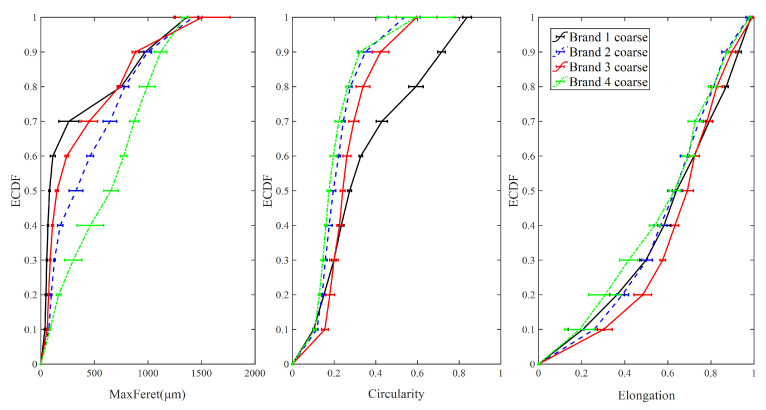
The empirical cumulative distribution function of shape factors for coarse particles.

**Figure 9 foods-09-01024-f009:**
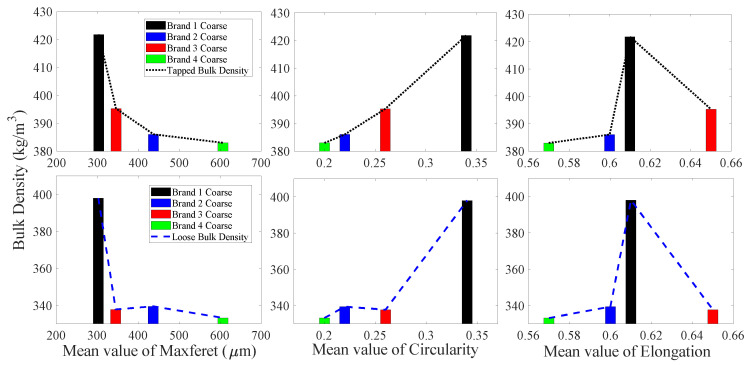
Tapped and loose bulk density from four different brands of instant whole milk powder samples versus mean values of shape factors for coarse particles.

**Figure 10 foods-09-01024-f010:**
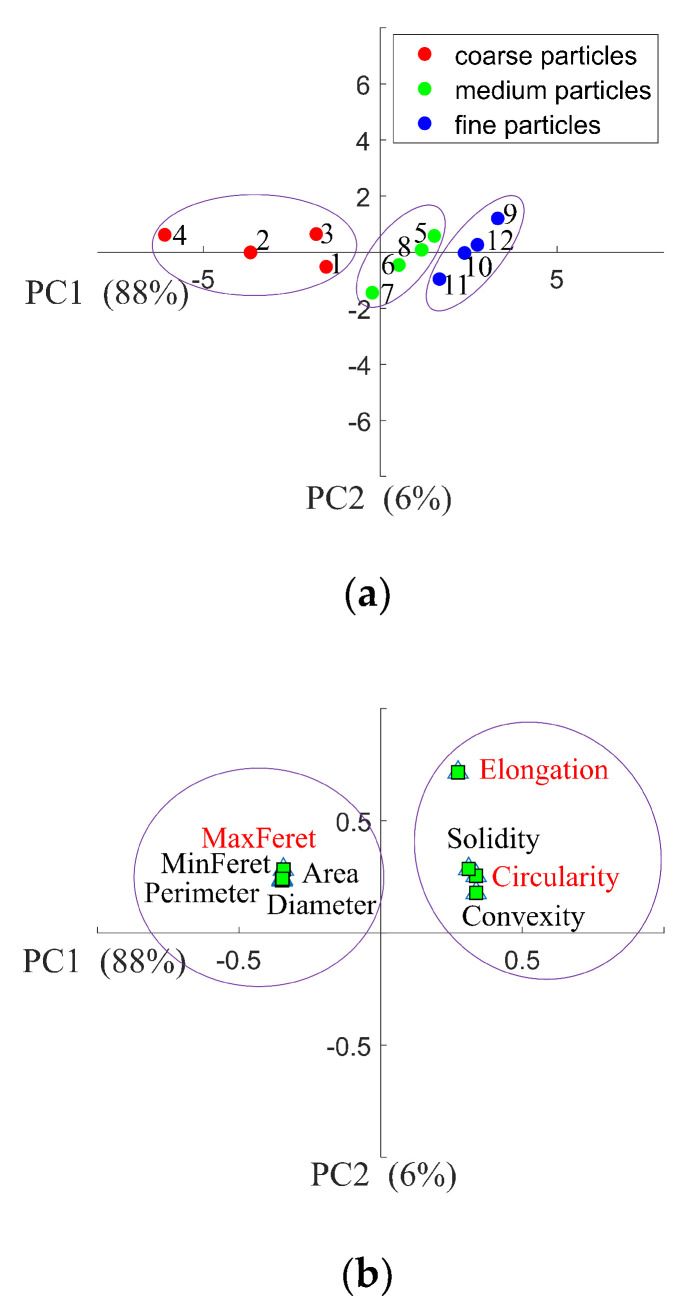
(**a**) Principal Component Analysis (PCA) scores plot of the first two principal components distinguishing the twelve instant whole milk powder samples from their bulk density; (**b**) PCA loadings plot of the first two principal components distinguishing the relationships between different shape factors.

**Figure 11 foods-09-01024-f011:**
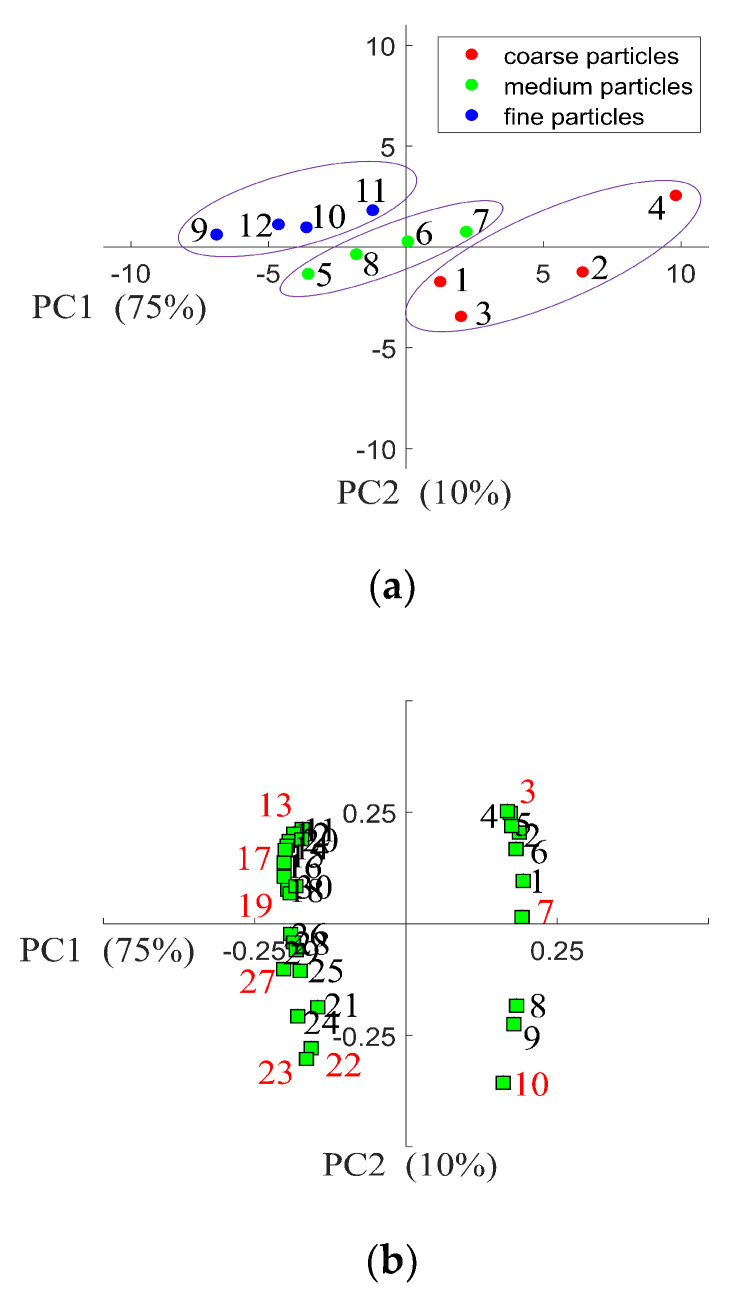
(**a**) PCA scores plot of the first two principal components allowing the discrimination of the twelve instant whole milk powder samples from their bulk density; (**b**) PCA loadings plot of the first two principal components allowing the determination of the relationships between different shape factor variables.

**Figure 12 foods-09-01024-f012:**
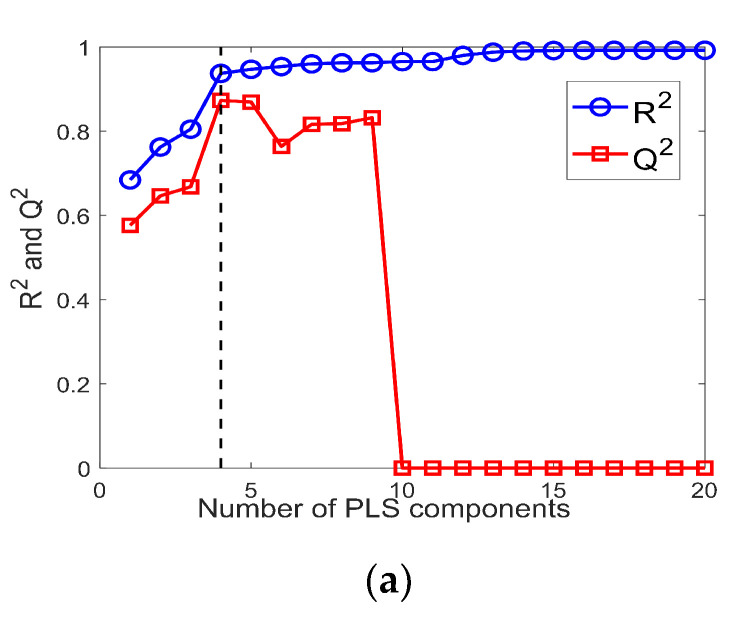
(**a**) R2 and Q2 versus the number of partial least squares (PLS) components for the PLS model of loose bulk density; (**b**) Predicted loose bulk density versus actual loose bulk density of IWMP samples.

**Figure 13 foods-09-01024-f013:**
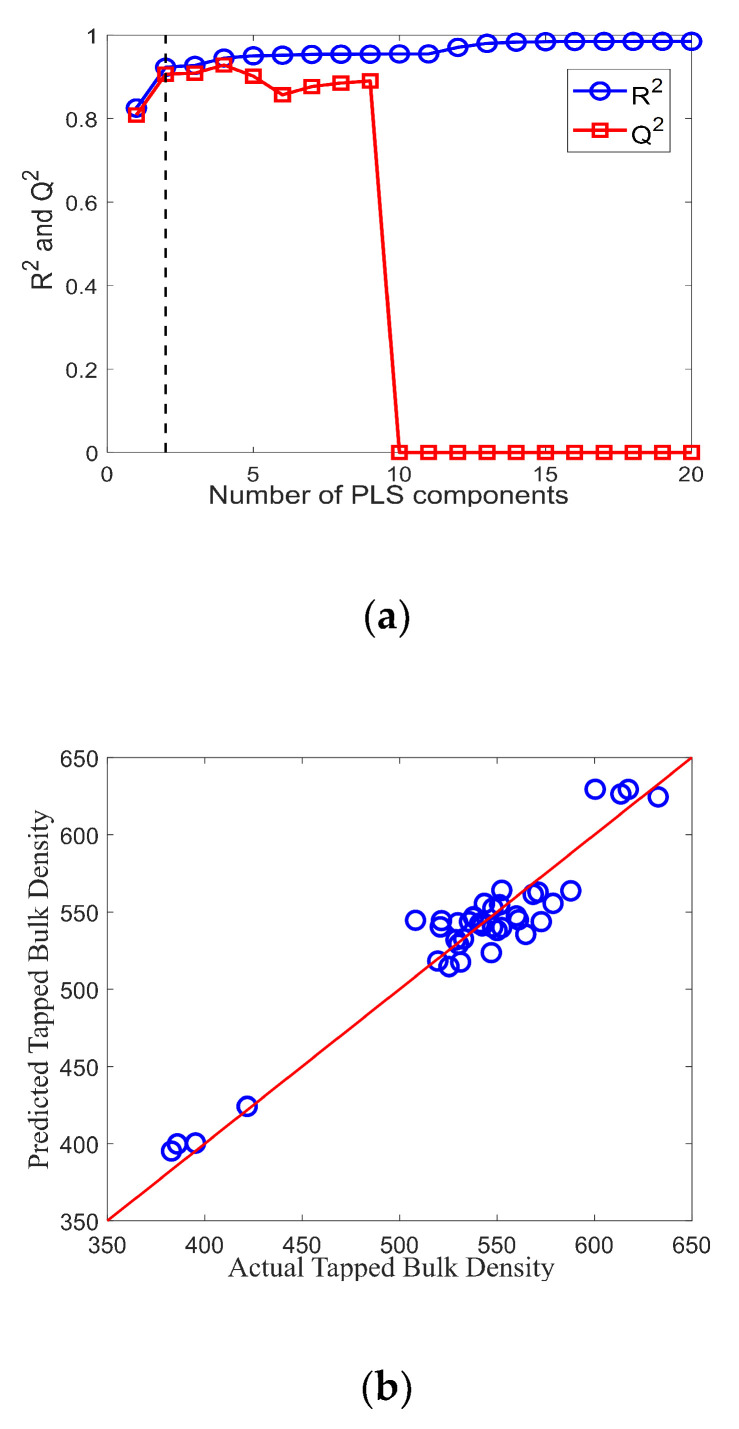
(**a**) R2 and Q2 versus the number of PLS components for the PLS model of tapped bulk density; (**b**) Predicted tapped bulk density versus actual tapped bulk density of IWMP samples.

**Figure 14 foods-09-01024-f014:**
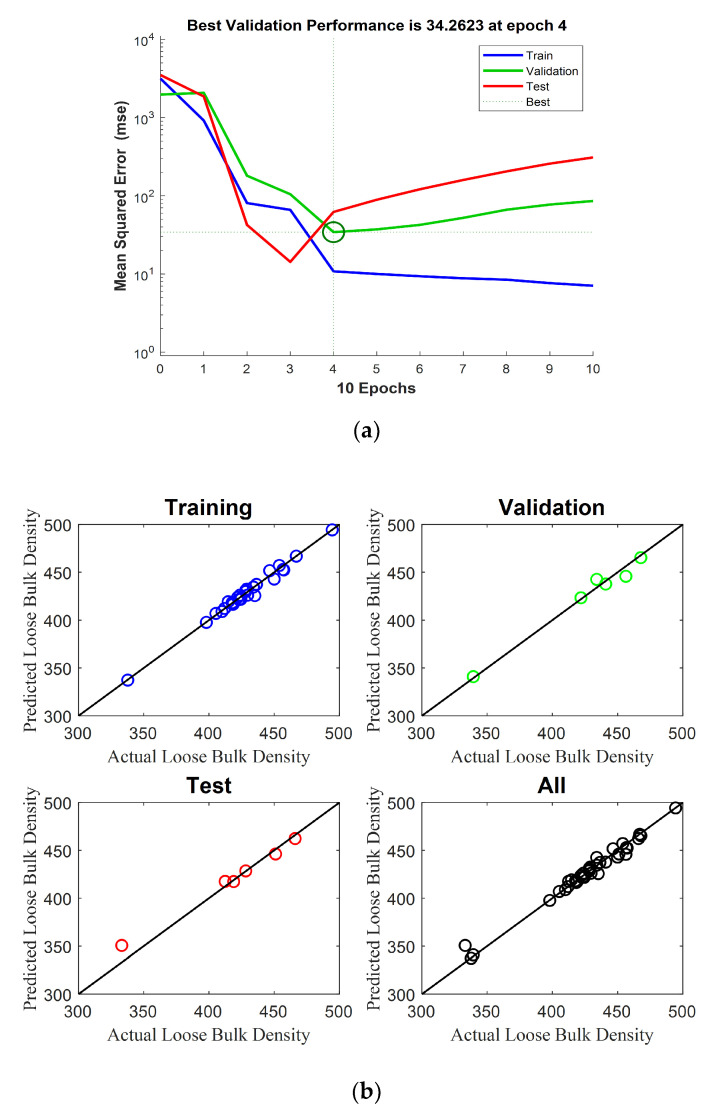
(**a**) Validation of the artificial neural networks (ANN) model of loose bulk density; (**b**) Performance of the ANN model of loose bulk density.

**Figure 15 foods-09-01024-f015:**
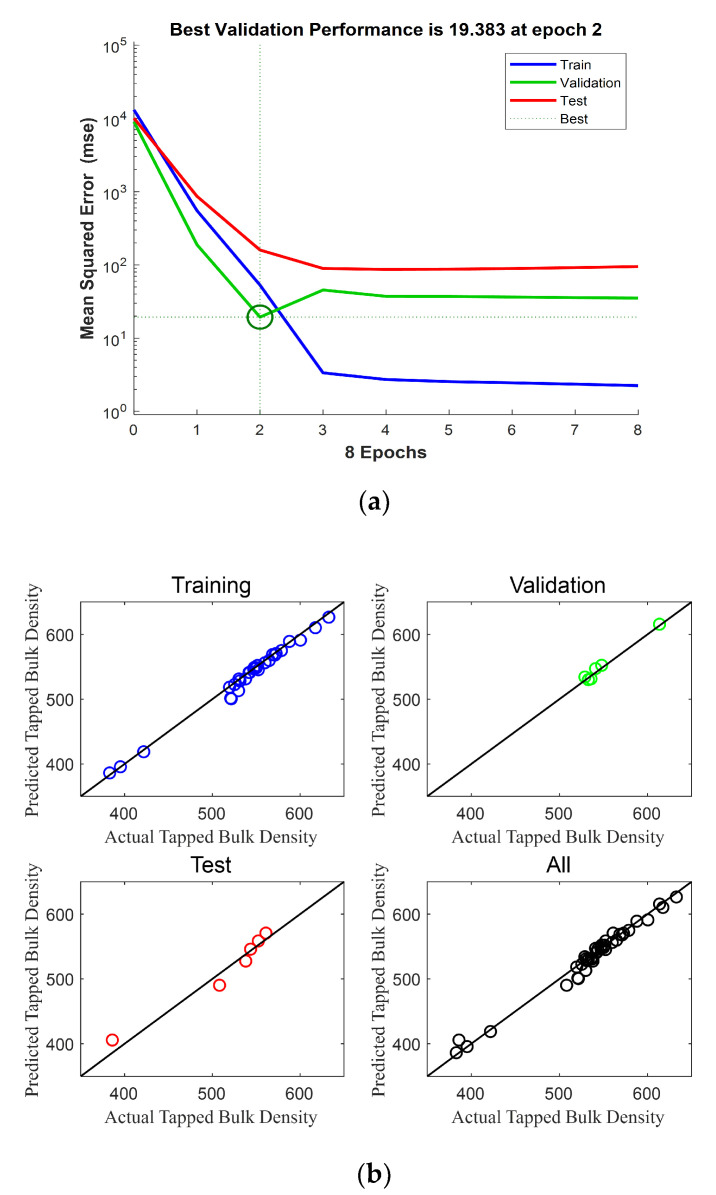
(**a**) Validation of the ANN model of tapped bulk density; (**b**) Performance of the ANN model of tapped bulk density.

**Table 1 foods-09-01024-t001:** Tapped and loose bulk density of coarse, medium, and fine particles for four brands of milk powders.

		Brand 1	Brand 2	Brand 3	Brand 4
Loose bulk density (kg/m^3^)	Coarse	397.99 ± 3.67	339.40 ± 0.43	337.76 ± 1.11	333.17 ± 1.05
	Medium	433.97 ± 8.17	417.79 ± 6.84	409.95 ± 4.22	422.88 ± 2.83
	Fine	494.41 ± 2.23	466.90 ± 5.36	440.84 ± 5.82	467.74 ± 7.62
Tapped bulk density (kg/m^3^)	Coarse	421.81 ± 4.76	386.03 ± 1.48	395.28 ± 0.95	382.98 ± 0.39
	Medium	529.88 ± 3.75	508.21 ± 2.01	521.54 ± 1.31	520.94 ± 0.40
	Fine	632.73 ± 3.12	613.60 ± 1.36	600.41 ± 3.20	617.45 ± 0.95

**Table 2 foods-09-01024-t002:** Tapped and loose bulk density of each sample class for the four different brands.

	Sample Class	Brand 1	Brand 2	Brand 3	Brand 4
Loose bulk density (kg/m^3^)	4	457.35 ± 8.25	424.42 ± 0.91	405.29 ± 3.53	412.36 ± 2.50
	5	466.00 ± 6.66	449.95 ± 1.74	423.75 ± 2.09	436.35 ± 1.86
	6	456.31 ± 6.82	429.58 ± 2.18	418.49 ± 1.38	421.91 ± 1.23
	7	453.90 ± 2.71	428.94 ± 1.26	418.29 ± 6.68	428.17 ± 2.33
	8	456.65 ± 5.33	434.00 ± 3.51	424.02 ± 1.36	428.14 ± 2.73
	9	451.02 ± 2.76	421.98 ± 5.31	411.96 ± 1.68	414.67 ± 1.68
	10	446.50 ± 1.84	428.07 ± 1.89	418.89 ± 3.10	435.14 ± 2.55
Tapped bulk density (kg/m^3^)	4	547.10 ± 1.58	531.36 ± 0.81	519.63 ± 1.51	525.39 ± 0.71
	5	587.91 ± 3.58	571.09 ± 1.97	552.46 ± 1.53	568.58 ± 0.74
	6	559.97 ± 2.96	537.92 ± 1.89	538.09 ± 1.24	535.88 ± 0.30
	7	572.73 ± 1.23	547.81 ± 2.18	542.48 ± 0.42	550.08 ± 1.18
	8	578.76 ± 1.11	551.39 ± 1.01	543.58 ± 0.55	547.97 ± 1.01
	9	564.82 ± 0.79	528.94 ± 0.81	532.86 ± 0.51	530.32 ± 0.41
	10	561.04 ± 1.05	541.04 ± 1.28	541.78 ± 2.26	552.40 ± 0.17

**Table 3 foods-09-01024-t003:** Results of the ANN model of loose bulk density.

	Samples	MSE	*R* ^2^
Training	28	10.80	0.98
Validation	6	34.26	0.98
Test	6	62.13	0.97
All	40	22.02	0.98

**Table 4 foods-09-01024-t004:** Results of the ANN model of tapped bulk density.

	Samples	MSE	*R* ^2^
Training	28	53.04	0.98
Validation	6	19.38	0.98
Test	6	159.71	0.96
All	40	63.99	0.98

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
