# Peer review of "Effects of Morphology on the Bulk Density of Instant Whole Milk Powder"

_foods, 2020, doi:10.3390/foods9081024_

Round 1

Reviewer 1 Report

This is a quite interesting subject with a very well structured introduction and detailed experimental part, but with many problems that are mainly related to the discussion of the research findings, which is extremely poor in terms that no explanations are given and no comparison with literature data are made.

Additionally, the report of the results is very monotonous and there are a lot of repetitions (especially in the case of subsections 3.1.1-3.13).

The conclusions need to be revised in order to provide the main findings (the first 4 lines are unnecessary).

Table 1: The inclusion of superscripts should be very helpful.

Author Response

General comments:

  1. This is a quite interesting subject with a very well structured introduction and detailed experimental part, but with many problems that are mainly related to the discussion of the research findings, which is extremely poor in terms that no explanations are given and no comparison with literature data are made.

The authors have taken this critique on board.  Further explanation has been added in the results and discussion section (line 437). ‘Compared to traditional time-consuming and labour-intensive off-line bulk density testing, using the PLS and ANN models developed in this work is potentially helpful to build on-line or at-line sensors, which is very necessary for industry 4.0. Additionally, Depree et al. [10] used process variables (pressure, flow, temperature, etc.) and online quality variables (moisture, fat, protein, etc.) to develop a partial least squares model for predicting the bulk density of IWMP. However, the precision (R2 is lower than 0.8) of using process variables to predict bulk density of IWMP is much poorer than the precision (R2 is about 0.94) of using morphology to predict bulk density of IWMP, which may be because that the bulk density of the agglomerated powders is mainly determined by particle shape [27].’

Specific comments:

  1. Additionally, the report of the results is very monotonous and there are a lot of repetitions (especially in the case of subsections 3.1.1-3.13).

The authors have noted this criticism and have altered the manuscript. Subsection 3.1.2 has been rewritten in the revised manuscript. ‘The ECDFs of maximum Feret diameter, elongation, and circularity are illustrated in Figure 6. It is noticeable that the ECDF curves of elongation and circularity for Brand 1 are well separated from the other three brands, and the tapped and loose bulk density of Brand 1’s medium sized particles are also significantly higher in comparison. Additionally, the values of circularity and elongation for Brand 3 are the highest while the loose bulk density of Brand 3 is the lowest. Consequently, in the medium particle size group, the loose bulk density of IWMP can be differentiated by elongation and circularity. However, the tapped bulk density of Brand 3 is higher than that for Brands 2 and 4, so, circularity and elongation cannot distinguish the tapped bulk density of IWMP in the medium particle size group. Furthermore, since the values of maximum Feret diameter of Brands 1-4 are similar, the maximum Feret diameter also cannot be used to distinguish bulk density of IWMP for the medium particle size fraction. Fig. 7 shows the mean values of shape factors (maximum Feret diameter, elongation, and circularity) for medium particles of different brands versus their tapped and loose bulk density. Similarly to the fine particles, with a decrease in maximum Feret diameter and an increase in circularity and elongation (more regular particle shape), the loose bulk density shows an overall upward trend (low interstitial air content). Additionally, with a decline in maximum Feret diameter, the tapped bulk density rises, but the trend of tapped bulk density versus the change of circularity and elongation is unclear.’ Additionally, some parts of the subsections 3.1.1 and 3.1.3 have been rewritten in the revised manuscript.

  1. The conclusions need to be revised in order to provide the main findings (the first 4 lines are unnecessary).

Acknowledged. The first 4 lines of the conclusions have been rewritten. ‘By investigating the relationship between morphology/particle size and bulk density, the current research is to develop an on-line or at-line sensor for bulk density measurement. The proposed methodology can improve plant efficiency and reduce operating costs as an alternative to traditional bulk density testing. It was found that…’

  1. Table 1: The inclusion of superscripts should be very helpful.

Thanks for your suggestion. Unfortunately, we are not sure what action the reviewer is recommending here and consequently Table 1 remains unchanged. We note that superscripts are already used to denote cubic metres as m3.  

Reviewer 2 Report

The authors have developed a rapid method for screening powders regarding bulk density by only using a microscope. If the following remarks are answered I recommend the publication of the article in Foods. 

The authors should provide some data concerning the moisture and oil content of the samples.

Line 235, 260, 281: Give possible explanation for the trends.

Since the comparison is between different brands and possible different manufacture lines the values of densities may differ due to other parameters as well, moisture, oil content. Some references should be mentioned.

The authors should specify that the trends regarding the densities are not correlated with the shape parameters you study. I suggest that the figures 5, 7 and 9 should changed into histograms so the values of the densities would be independent between the samples.  

Author Response

General comments:

The authors have developed a rapid method for screening powders regarding bulk density by only using a microscope. If the following remarks are answered I recommend the publication of the article in Foods.

Specific comments:

  1. The authors should provide some data concerning the moisture and oil content of the samples.

Thanks for pointing this out. Some sentences have been added in subsection 2.1.1 for clarification. ‘Additionally, although some other properties may affect the bulk density of IWMP, e.g. moisture and oil content, the bulk density of IWMP is mainly determined by particle shape and the amount of interstitial air [27]. And the aim of this work is to develop the on-line or at-line sensors to predict the bulk density of IWMP, so the effects of other properties on the bulk density of IWMP will be studied in the future.’

  1. Line 235, 260, 281: Give possible explanation for the trends.

A possible explanation for the trends has been added to the revised manuscript. ‘As circularity and elongation increase, the bulk density also increases, while as maximum Feret diameter increases, the bulk density decreases. This may be because that the milk powders with a more regular particle shape will cause low interstitial air content, which will lead to a high bulk density.’ ‘Similarly to the fine particles, with a decrease in maximum Feret diameter and an increase in circularity and elongation (more regular particle shape), the loose bulk density shows an overall upward trend (low interstitial air content).’ ‘It is clear that as maximum Feret diameter increases and the circularity decreases (more irregular particle shape), the bulk density decreases (high interstitial air content).’

  1. Since the comparison is between different brands and possible different manufacture lines the values of densities may differ due to other parameters as well, moisture, oil content. Some references should be mentioned

Thanks for pointing this out. A sentence has been added in subsection 3.1.3 for clarification. ‘Furthermore, although the moisture and oil content may affect the bulk density of IWMP [3], the moisture and oil content are controlled very tightly during milk powder production [27], so the differences in moisture and oil content between four different brands of IWMP should not be appreciable.’

  1. The authors should specify that the trends regarding the densities are not correlated with the shape parameters you study. I suggest that the figures 5, 7 and 9 should changed into histograms so the values of the densities would be independent between the samples.

The authors agree with the reviewer. The figures 5, 7 and 9 have changed into histograms.

Reviewer 3 Report

This paper presents the relationship between morphology and bulk density of instant whole milk powder and developed models for predicting bulk density of milk powder.

Line 13-16: ‘Bulk density… is affected by other physical properties, e.g. morphology and particle size.’ It's exactly the opposite; bulk density of powder mainly depends on his morphology and particle size.

Line 40-42: ‘…a higher bulk density is associated with better flowability [18,27].’ This statement hasn’t been confirmed by other researchers. Juliano et al. [18] didn’t mention about it. What about powder and its agglomerate (example whole milk powder and instant whole milk powder)? Which one will be have higher bulk density? And better flowability?

Line 64-65: ‘Since IWMP is mostly agglomerated particles [10], the IWMP’s bulk density is determined by the particle shape.’ Please give references (min. 2) for this fact.

Line 99-100: ‘150 g samples of milk powder were shaken for 30 minutes ….’ Sieving time was too long and could cause crushing, grinding of agglomerates. Which caused the particle size distribution to shift towards smaller values. It is recommended to sieve 100 g of agglomerate within 10 minutes.

Line 388-341 (Title of Figure 11): ‘….instant whole milk powder samples from their dispersibility….’ Dispersibility? Not measured; no methodology in section 2.

Line 342: ‘Prediction of the dispersibility’ Dispersibility? Not measured; no methodology in section 2.

Author Response

General comments:

This paper presents the relationship between morphology and bulk density of instant whole milk powder and developed models for predicting bulk density of milk powder.

Specific comments:

  1. Line 13-16: ‘Bulk density… is affected by other physical properties, e.g. morphology and particle size.’ It's exactly the opposite; bulk density of powder mainly depends on his morphology and particle size.

Thanks for pointing this out. This sentence has been rewritten in the revised manuscript. ‘Bulk density, which directly determines the packing cost and transportation cost of milk powder, is one of the most important functional properties of IWMP, and it is mainly affected by physical properties, e.g. morphology and particle size.’

  1. Line 40-42: ‘…a higher bulk density is associated with better flowability [18,27].’ This statement hasn’t been confirmed by other researchers. Juliano et al. [18] didn’t mention about it. What about powder and its agglomerate (example whole milk powder and instant whole milk powder)? Which one will be have higher bulk density? And better flowability?

Thanks for pointing this out. This sentence has been rewritten in the revised manuscript (line 41). ‘Furthermore, bulk density is an important parameter describing the flowability of milk powder [18]. Since the agglomeration increases the particle size, it significantly reduces the bulk density of milk powder [3,27]. Therefore, the bulk density of instant whole milk powder is generally lower than the bulk density of regular whole milk powder. Additionally, since the free-flowing property is improved by agglomeration [27], regular whole milk powder always has a poorer flowability compared with the agglomerated/instant whole powder.’

  1. Line 64-65: ‘Since IWMP is mostly agglomerated particles [10], the IWMP’s bulk density is determined by the particle shape.’ Please give references (min. 2) for this fact.

Thanks for pointing this out, this sentence has been rewritten for clarification. ‘Since the non-agglomerated particles are recycled from the spray dryer chamber and the rest of which are removed in the fluid bed, IWMP is mostly agglomerated particles [10,27]. Therefore, the IWMP’s bulk density is determined by the particle shape.’

  1. Line 99-100: ‘150 g samples of milk powder were shaken for 30 minutes ….’ Sieving time was too long and could cause crushing, grinding of agglomerates. Which caused the particle size distribution to shift towards smaller values. It is recommended to sieve 100 g of agglomerate within 10 minutes.

Thanks for pointing this out, this sentence has been rewritten for clarification. ‘150 g samples of milk powder were shaken for 30 minutes before the powders were transferred to foil bags to ensure stable moisture content. To minimize the breakdown of agglomerates, the sieve shaker used was a vibratory type and the amplitude selected was low. It is worth mentioning that the milk powder samples that were measured came from packer, which have already gone through significant dense phase vacuum transport. Therefore, only the most robust powder is left behind at this point, everything else is crushed during transport [6].’

  1. Line 388-341 (Title of Figure 11): ‘….instant whole milk powder samples from their dispersibility….’ Dispersibility? Not measured; no methodology in section 2.

Thanks for pointing this out. The word ‘dispersibility’ was replaced with the words ‘bulk density’.

  1. Line 342: ‘Prediction of the dispersibility’ Dispersibility? Not measured; no methodology in section 2.

Thanks for pointing this out. The word ‘dispersibility’ was replaced by the words ‘bulk density’.

Round 2

Reviewer 1 Report

The manuscript has been corrected according to the suggestions.

Reviewer 2 Report

The authors have answered the reviewers' comments in a satisfactory level, so I recommend the publication of the article.